# Cropland Mapping Using Sentinel-1 Data in the Southern Part of the Russian Far East

**DOI:** 10.3390/s23187902

**Published:** 2023-09-15

**Authors:** Konstantin Dubrovin, Alexey Stepanov, Andrey Verkhoturov

**Affiliations:** 1Computing Center Far Eastern Branch of the Russian Academy of Sciences, 680000 Khabarovsk, Russia; 2Far Eastern Agriculture Research Institute, Vostochnoe, 680521 Khabarovsk, Russia; 3Khabarovsk Federal Research Center of the Far Eastern Branch of the Russian Academy of Sciences, 680000 Khabarovsk, Russia

**Keywords:** remote sensing, crop identification, time series classification, SAR data, DpRVI, machine learning

## Abstract

Crop identification is one of the most important tasks in digital farming. The use of remote sensing data makes it possible to clarify the boundaries of fields and identify fallow land. This study considered the possibility of using the seasonal variation in the Dual-polarization Radar Vegetation Index (DpRVI), which was calculated based on data acquired by the Sentinel-1B satellite between May and October 2021, as the main characteristic. Radar images of the Khabarovskiy District of the Khabarovsk Territory, as well as those of the Arkharinskiy, Ivanovskiy, and Oktyabrskiy districts in the Amur Region (Russian Far East), were obtained and processed. The identifiable classes were soybean and oat crops, as well as fallow land. Classification was carried out using the Support Vector Machines, Quadratic Discriminant Analysis (QDA), and Random Forest (RF) algorithms. The training (848 ha) and test (364 ha) samples were located in Khabarovskiy District. The best overall accuracy on the test set (82.0%) was achieved using RF. Classification accuracy at the field level was 79%. When using the QDA classifier on cropland in the Amur Region (2324 ha), the overall classification accuracy was 83.1% (F1 was 0.86 for soybean, 0.84 for fallow, and 0.79 for oat). Application of the Radar Vegetation Index (RVI) and VV/VH ratio enabled an overall classification accuracy in the Amur region of 74.9% and 74.6%, respectively. Thus, using DpRVI allowed us to achieve greater performance compared to other SAR data, and it can be used to identify crops in the south of the Far East and serve as the basis for the automatic classification of cropland.

## 1. Introduction

The transition to digital agriculture can be achieved by addressing the challenges associated with the automated recognition of crops growing in a given area [1]. In recent years, research at both the national and global scale has focused on creating regional cropland masks of different regions, as well as masks of individual crops [2,3]. Machine learning (ML) methods are actively used to solve these problems. Typically, input data for these ML models are seasonal time series of various indices (especially optical) that are obtained after processing images. This is because time series usually provide higher accuracy in mapping than an index calculated from a single image [4,5,6].

Typically, to identify agricultural crops, time series of optical vegetation indices (VI) and traditional ML classifiers, such as decision trees, Support Vector Machines (SVM), and Random Forest (RF) algorithms are used. An example of this is a study conducted in the Tashkent province, Uzbekistan, in which researchers classified several crops (cotton, wheat, rice, other crops) using a series of monthly Normalized Difference Vegetation Index (NDVI), Enhanced Vegetation Index (EVI), and Normalized Difference Water Index (NDWI1 and NDWI2) composites, based on Landsat-8 and Sentinel-2 imagery [7]. The highest accuracy was 90% (Kappa = 0.88) when using EVI composites derived from Landsat-8 data. Hu et al. performed retrospective restoration of crop rotation using the RF algorithm [8]. They trained the classifier on percentile monthly Sentinel-2 composites in 2020 to map cropland in 2019. The possibility of successful classification 2 months before the end of the growing season was shown. The overall accuracy of retrospective forecasting was 92%. In this study, monthly bands and VI time series were used. The use of a large number of correlating parameters (in the model [8], 90 parameters were included) in the classification model requires a large number of calculations and may be excessive.

A serious disadvantage of optical data used in the discussed works is the strong dependence of satellite data quality on weather conditions. Clouds, shadows from clouds, and the presence of aerosols in the atmosphere may significantly reduce the potential of optical sensors. A large number of cloudy images leads to substantial gaps in VI time series built using optical imagery.

Some researchers have used the approximation of the seasonal course of VI to bridge gaps in remote sensing data. Polynomials, Gauss function, and the double logistic function [9,10,11] are most often used as approximate functions. In Uruguay, scientists developed a soybean growth model and predicted the NDVI maximum using polynomials and the double logistic function for fields with an area of more than 250 hectares [12]. In the Samara region of Russia, a group of researchers [13] approximated the seasonal course of NDVI using a piecewise function, Fourier series, and cubic splines.

SAR (Synthetic Aperture Radar) data, independent of lighting and weather conditions, is a reliable alternative for long-term monitoring of crop rotation. Sentinel-1A/B satellites sensors allow the observation of a significant part of the Earth and receive double polarization data every 12 days (every 6 days when using both satellites). This is sufficient to build continuous time series with a sufficiently large number of observations for the season. Therefore, radar data obtained from Sentinel-1 has recently been actively used for crop mapping and the identification of abandoned cropland. For instance, Qiu et al. [14] used time series of VV polarization to detect woody abandoned cropland in China. However, to determine grass fallows, single polarization was not enough, and the authors additionally used optical data. In the majority of studies, scientists have used either the VH or VV polarizations, or even their ratio (VH/VV). For example, Song et al. [15] determined the proportion of pixels of soybean and corn (three classes per crop: small, middle, and large) in experimental polygons throughout the United States. Sentinel-1 VV, VH, and VV/VH time series for one agricultural season were used. The use of three predictors in a decision tree classifier achieved an accuracy greater than 94% for both crops. Arias et al. [16] attempted to identify 14 crops based on Sentinel-1 radar data (three sets of polarizations) at the level of the province of Navarra in Spain, as well as that of individual municipalities. The accuracy of the classification using Sentinel-1 did not exceed 70%. Attempts have also been made to merge radar and optical data for crop mapping [17,18], where Landsat-8 (or Sentinel-2) and Sentinel-1 imagery were used as initial data and the RF algorithm was used as a classification method. In both of these studies, significant sparsity in the data was observed, and linear interpolation greatly decreased the accuracy of the classification.

Time series for several years make it possible to observe crop rotation. For example, a group of scientists [19] used the Dynamic Time Warping method and linear interpolation to equalize data for two years when classifying nine crops in China based on the combination of polarizations.

Despite the frequent and successful use of individual Sentinel-1 polarizations for crop mapping, the use of radar VIs (by analogy with well-known optical VIs) appears to be more promising because they are constructed for monitoring vegetation. The Radar Vegetation Index (RVI) is the most common radar index [20]. Evidence suggests that this index is highly sensitive to the dynamics of plant growth; therefore, it can likely be used to assess crop phenology and the characteristics of vegetation and perform classification [17,21]. Bickensdorfer et al. [22] classified croplands in Germany using RVI, VV, VH and the ratio of VV/VH monthly composites. Classification accuracy while using four time series (12 values) was 65%. To increase the accuracy to 80%, the authors had to additionally calculate the time series of several vegetation indices based on Sentinel-1 and Sentinel-2 data, resolve inconsistencies in the spatial resolution, restore data, and add temperature, precipitation, and humidity variables to the model. Gella et al. [23] used Dual Polarization SAR Vegetation Index (DPSVI) and Modified Radar Vegetation Index (MRVI) time series to identify eight agricultural crops in the Netherlands. The use of such VIs has been found to improve overall accuracy.

The Dual-polarization Radar Vegetation Index (DpRVI) is based on the transformations of complex polarimetric radar Level-1 Single Look Complex (SLC) data [24]. Processing is carried out by applying one of the polarimetric decomposition methods [25,26,27]. At the same time, a comprehensive polarimetric response of the signal from the object to the components, which characterizes the contribution of a particular radar signal (single, double, or volumetric), is obtained. Information about scattering is characterized by the degree of polarization and the measure of the dominant scattering mechanism. Based on these calculated indicators, as shown by [24], DpRVI becomes more sensitive to crop growth and is used as a relatively simple and physically interpretable vegetation descriptor.

Croplands in different regions have significant differences in their spectral characteristics, which leads to inaccurate forecasts even if the classifier is well-trained [28]. In particular, a number of researchers, including those from Russia, have faced the problem of incorrect results from the model when it was transferred to regions where crops had a different phenology and the temporal spectral profile of the crops was changing [29,30].

In recent years, the challenges associated with creating a cropland mask have been addressed at the Space Research Institute in Russia. Land classification is carried out using NDVI composites on cropland (spring and winter crops separately), forest, grassy vegetation, swamps, and tundra. Miklashevich et al. [30] achieved high accuracy in land mapping for the European regions of Russia. The disadvantage of large-scale maps is the significantly uneven quality of the classification between regions (in particular, Miklashevich et al. [30] noted an increase in error in the Russian Far East). It can also be noted that although most of these maps are aimed at determining the land types and the creation of a cropland mask, they do not provide any information about crop rotation on the allocated lands. Moreover, since different regions are characterized by different sets of agricultural crops and the variety of climatic conditions leads to variation in phenology, the task of separating crops based on VI can be extremely difficult even if the classifier was well performed in one region. Therefore, the most productive approach is the development of accurate local maps indicating the growing crops within the region or in a few regions characterized by similar climatic conditions and agricultural practices.

The main crop in the Russian Far East is soybean. Soybean occupies 60–90% of the cropland area in various municipalities [31]. It should be noted that the increase in soybean acreage is not as much due to the commissioning of new lands to crop rotation but due to repeated sowing. This is unacceptable in terms of the recommended crop rotations, where its specific share should not exceed 50%. The placement of soybean in crop rotation patterns contributes to an increase in yield by an average of 36.4% compared to that in the case of permanent cultivation. However, repeated sowing of this crop in crop rotations reduces its yield by 27.2% [32]. To ensure the optimal use of the cropland, soybean is rotated with grain crops, including oat [33,34].

However, there are a lot of abandoned croplands in the Russian Far East. Currently, the total fallow area in Russia is up to 40 million ha. According to the data obtained from the agricultural census conducted in 2016, nearly 17.7 million ha of fallow land are owned by agricultural enterprises yet remain unclaimed by them. According to the Ministry of Agriculture of the Russian Federation, this area was approximately 15.3 million ha in 2019 [35]. Such lands include both bushy and forested land, as well as swampy fallow land. Defining fallow lands and introducing them into the crop rotation system are among the most important tasks for the agricultural sector in the country.

Thus, the main objective of this study is to identify the main crop of the region (soybean) used in crop rotation, as well as oat and fallow lands, by developing cropland classifiers for the Khabarovsk District based on different DpRVI time series and ML methods. Further, it aims to assess the possibility of using best classifiers for other municipalities in the Russian Far East, and compare the differences in the accuracy of determining these classes across these regions.

## 2. Materials and Methods

### 2.1. Study Areas

The municipal districts of the southern part of the Russian Far East in the Amur River basin were chosen as the study areas. Study area 1 (SA1) is the cropland of the Khabarovskiy District, situated on the right bank of the Amur River in the vicinity of the city of Khabarovsk. It is located between 48.31° N and 48.64° N and between 134.81° E and 135.57° E of the Greenwich meridian (Figure 1). It is characterized by favorable climate and soil conditions, which makes it possible to grow grain and leguminous crops.

Three districts (Arkharinskiy, Ivanovskiy, and Oktyabrskiy) are located on the territory of the Zeya-Bureinskaya Plain (Amur Region). About 75% of the territory of the Zeya-Bureya Plain of the Amur Region has been converted into agricultural landscapes. The total area of cropland in the territory of the Zeya-Bureya plain is 1.32 million ha. Cropland is the most important type of agricultural land and the main wealth of the region. Study area 2 (SA2) is located between 49.28° N and 50.61° N, 127.76° E and 130.05° E of the Greenwich meridian (Figure 2). Relatively high temperatures and the length of the vegetation season make it possible to grow good and stable yields of cereals, industrial, and other crops [36].

The study areas are characterized by a monsoonal climate with cold snowy winters (the average January temperature varies from −25 °C in study area [SA] 2 to −20 °C in SA1) and humid warm summers (the average July temperature in the study areas exceeds 21 °C). The average annual rainfall is at the level of 600–700 mm per year.

The main agricultural crop in the southern part of the Russian Far East is soybean; in 2021, soybean occupied 55% (9381 ha) in the Khabarovskiy District, 79% (28,596 hectares) in the Arkharinskiy District, 76% (82,425 ha) in the Ivanovskiy District, and 79% in the Oktyabrsky district (78,899 ha) of the total area of arable land. The land occupied by oats is 7% (1232 ha), 5% (1851 ha), less than 1% (865 ha), and just over 1% (1380 ha) in the Khabarovskiy District, Arkharinsky District, Ivanovsky District, and Oktyabrskiy District, respectively.

### 2.2. Data

#### 2.2.1. DpRVI Calculation

To calculate DpRVI, Sentinel-1 Level-1 SLC images were acquired for SA1 from one track (one scene with relative orbit number 90), but for SA2 from two tracks (two scenes with relative orbit number 105 and one scene with relative orbit number 134), as shown in Table 1. The data were obtained from the Alaska Satellite Facility Distributed Active Archive Center (https://search.asf.alaska.edu/ (accessed on 14 January 2023), which contains modified Copernicus Sentinel data from 2015, processed by the European Space Agency. The *Geopandas* package was used for raster data handling. Two scenes with track number 105 between 2 May and 29 October 2021 were selected. The scene with track number 134 was acquired between 4 May and 31 October 2021, and the scene with track number 90 was acquired between 1 May and 28 October 2021. These data were used in the processing Terrain Observation with Progressive Scans SAR (TOPSAR) mode Interferometric Wide (IW) swath products. The swath length is 250 km and the spatial resolution is 5 × 20 m in this configuration. Swath in the IW mode is divided into three sub-swaths (IW1, IW2, and IW3). Each sub-swath consisted of 9 bursts in the azimuth direction.

SA1 is within one sub-swath of scene 90. SA2 is located in three zones, marked with red rectangles with a dash-dotted line in Figure 3. Fields that are in the western part are included in the zone of intersection of two scenes from the Sentinel-1B satellite, while in the eastern part they are concentrated in one scene. Thus, pre-processing of the radar image time series data was carried out using two different workflows, with standard correction steps as follows [37]: (a) assembly of two slices (relative orbit number 105 of IW1); (b) single sub-swath processing (relative orbit number 134 of IW3).

All workflows were processed using the Graph Processing Framework (GPF) on the ESA Sentinel Application Platform (SNAP) v.8.0 (http://step.esa.int/main/ (accessed on 14 January 2023)). This allows the user to create graphs and set up chains for batch processing, as shown in Figure 4. Pairs of SAR images for the same date were fed as input.

In Module 1, the operator S1 TOPS Split was used to select sub-swaths and bursts depending on the location of the study area. Slice Assembly combined parts of different scenes into a single product. Further, the precise orbit state vector files, which contained accurate information about the satellite position and speed, were downloaded and applied. Then, radiometric calibration was performed, during which it was important to keep the data in a complex-valued format because complex values were needed to calculate the covariance matrix for dual-polarization C2 in the subsequent steps. Next, in Module 2, the S-1 TOPS Deburst operator was applied to produce a single SLC image. These image subsets were multi-looked with factors of 3 and 1 in the range and azimuth directions, respectively, to generate a square pixel. The spatial resolution of the output was 10 m. Then, for each date for the entire data stack, a dual-polarization 2 × 2 covariance matrix C2 was formed, as shown in Equation (1):(1)$$C_{2}=\left[\begin{array}{cc} C_{11} & C_{12}\\ C_{21} & C_{22} \end{array}\right]=\left[\begin{array}{cc} \langle {\left|S_{VV}\right|}^{2}\rangle & \langle \left|S_{VV}S_{VH}^{*}\right|\rangle \\ \langle \left|S_{VH}S_{VV}^{*}\right|\rangle & \langle {\left|S_{VH}\right|}^{2}\rangle \end{array}\right],$$
where * indicates complex conjugation and 〈〉 indicates the spatial average over a moving window.

All elements of the matrix *C_2_* were additionally subjected to the noise mitigation procedure using the refined Lee adaptive filter [38]. The elements of the matrix were complex values containing all of the information about the polarimetric scattering properties. The polarimetric decomposition technique was then used to obtain the parameters of the state of polarization of an electromagnetic wave, which was characterized by the degree of polarization *m* [24,39] and the measure of the dominant scattering mechanism *β* of the reflecting target as follows:(2)$$m=\sqrt{\frac{4\left|C_{2}\right|}{{\left(Tr(C_{2})\right)}^{2}}},\;\beta \;=\;\frac{\lambda_{1}}{\lambda_{1}+\lambda_{2}},$$
where *Tr* represents the sum of the diagonal elements of the matrix, || represents the determinant of a matrix, *m* is the degree of polarization (0 ≤ *m* ≤ 1), which is defined as the ratio of the average intensity of the polarized portion of the wave to the average total intensity of the wave, and *β* refers to the measure of the dominant scattering mechanism, which is determined based on the decomposition of the matrix *C_2_* into two non-negative eigenvalues (*λ*_1_ ≥ *λ*_2_ ≥ 0), as follows:(3)$$C_{2}=U_{2}\cdot \Sigma \cdot U_{2}^{-1},\;where\;\Sigma =\left[\begin{array}{cc} \lambda_{1} & 0\\ 0 & \lambda_{2} \end{array}\right].$$

Further, the DpRVI index was calculated for each date using Equation (4):(4)$$DpRVI=1-m\beta =1-\sqrt{\frac{4\left|C_{2}\right|}{{\left(Tr(C_{2})\right)}^{2}}}\cdot \frac{\lambda_{1}}{\lambda_{1}+\lambda_{2}}.$$

In Module 3, all Sentinel-1 images were co-registered as a stack using the Sentinel-1 Back Geocoding operator and subsetted based on the area of interest. Following this, SAR stack data were geocoded to a WGS84/UTM zone 52N projected coordinate system using the Range Doppler Terrain correction operator to yield a DpRVI time-series product in GeoTiff format.

For single sub-swath processing (e.g., IW3), Module 1 was modified as shown in Figure 5. Since all studied fields were in one IW3 for track 134, Modules 2 and 3 remained the same as for the first workflow.

It must be noted that when several sub-swaths (e.g., IW1 and IW2) are processed, Module 2 can be modified, as shown in Figure 6. The S-1 Back Geocoding and S-1 TOPS Merge operators can be added to merge separated products of different sub-swaths into a single image.

#### 2.2.2. Ground Reference Data

Labeled DpRVI data seasonal series for pixels of agricultural fields of the Khabarovskiy District were divided into training data to build a model and test dataset to evaluate the quality of the model. Information on crop rotation, obtained from ground-based observations, was additionally verified by expert visual interpretation of satellite images. Overall, pixels from 80 fields were divided into three classes: soybean, oat, and fallow. Pixels of 40 fields were used to build a model, pixels of another 40 sites were used to evaluate model accuracy.

Table 2 provides information on the number of fields and areas occupied by the different crops. It can be seen that half of the fields (20) from the training sample in 2021 were occupied by oat; however, soybean was planted in more large fields, and thus the total area covered by soybean exceeded the area covered by oat. The share of fallow land in the studied fields of the Khabarovskiy District exceeded 35%.

Labeled DpRVI time series for pixels of agricultural fields in the districts of the Amur Region were used to evaluate the quality of the model when transferred to a neighboring region. To form a test sample, we selected abandoned fields, as well as fields with soybean and oat, in each of the three districts of the Amur Region: Arkharinskiy, Ivanovskiy, and Oktyabrskiy. To mark up the test data, we used information on crop rotation for the fields from the Unified Federal Information System of Agricultural Lands (https://efis.mcx.ru/landing (accessed on 26 December 2022)). During the preliminary visual interpretation of satellite images, we discovered that some of the fields were in the flood zone, while for other fields, information about the growing crop was deemed unreliable.

Overall, 29 fields were used to form an Amur region sample. Table 3 provides information on the number of fields and areas of land included in this sample.

### 2.3. Datasets

As shown in Table 1, there was no Sentinel-1 image for track 105 on 18 August 2021 and for track 134 on 3 July 2021. DpRVI time series were approximated using a cubic polynomial to restore gaps. The dates of observations converted to DOY (day of the year) format were considered an independent variable. DpRVI data values were used as the dependent variable. The polynomial coefficients were determined for each time series and then the function values were calculated at the points of interest: 230 (18 August) for track 105 and 184 (3 July) for track 134. Thus, after data recovery, the time series included 16 observations.

Despite the fact that DpRVI data are practically independent of atmospheric phenomena, anomalous values of this index can be observed in some parts of the fields. To minimize the influence of such noises, “2σ-filtering” was performed. For each field and for each date, the mean value and standard deviation (σ) were calculated for DpRVI. Pixels with more than half of the values of the time series that did not lie in the 2σ-interval were considered anomalous and were not included in the training set.

After removing anomalous pixels from the sample, a training dataset was formed, which consisted of 71,153 DpRVI time series. The test dataset for Khabarovsk District was not filtered, and its size was 30,550 time series for DpRVI. Then, DpRVI time series for Amur region fields were calculated and the Amur region dataset was formed. The Amur region set was also not filtered and included 203,120 time series. The distribution of time series between classes in both samples is presented in Table 4.

### 2.4. ML Methods

Classification was performed using the Python *scikit-learn* package. This package contains different machine learning approaches for supervised classification, such as tree algorithms, discriminant analysis, Bayes classifiers, nearest-neighbor techniques, ensemble classifiers, and neural networks. This package contains out-of-the-box implementations and does not require thorough tuning. We selected three machine learning algorithms for cropland classification: SVM, RF, and Quadratic Discriminant Analysis (QDA). The *GridSearchCV* method *(scikit-learn)* was applied to optimize SVM and RF hyperparameters during cross-validation. Random state controlled the pseudo random number generation for shuffling the data and reproducible output across multiple function calls.

#### 2.4.1. Support Vector Machines

SVM is a set of supervised learning methods used for classification, regression, and outlier detection. The advantages of SVMs include effectiveness in high dimensional spaces, memory efficiency (they use a subset of training points in the decision function, which are called support vectors), and versatility (different kernel functions can be specified for the decision function). The objective of the SVM is to find a hyperplane with the maximum margin (the maximum distance between data points of classes) in an N-dimensional space (N represents the number of features). LinearSVC is a fast implementation of Support Vector Classification in *scikit-learn* for the case of a linear kernel. This classification method is widely used in remote sensing for crop type mapping [40,41,42].

#### 2.4.2. Random Forest

RF [43] is a meta estimator that fits a number of decision tree classifiers on various sub-samples of the dataset and makes final decisions using majority votes to improve the predictive accuracy and control over-fitting. The features are always randomly permuted at each split. Therefore, the best split may vary even with the same training data. To obtain deterministic behavior during fitting, a random state has been fixed. Evidence suggests that RF is effective for crop mapping [18,22,44]. Moreover, it is known for performing particularly well and efficiently on large input datasets with many different features. Another advantage of RF is its high accuracy and robustness to outliers and noise [40].

#### 2.4.3. Quadratic Discriminant Analysis

QDA is the classic classifier, with a quadratic decision surface. It is attractive because it has closed-form solutions that can be computed easily, is inherently multiclass, has been proven to work well in practice [45], and has no hyperparameters to tune. QDA can be derived from simple probabilistic models that model the conditional distribution of the data for each class [46]. More specifically, QDA models data distribution as a multivariate Gaussian distribution. To train a classifier, the fitting function estimates the parameters of a Gaussian distribution for each class.

### 2.5. Classification Performance Evaluation

The accuracy metrics for each class, namely, overall accuracy (*OA*) and F1 metrics, were calculated based on a confusion matrix. A confusion matrix is a square matrix of dimensions *n × n* (where *n* is the number of classes). In the confusion matrix, each row represents an actual class, and each column represents a predicted class. Each element of the matrix *X_ij_* denotes the number of time series of class *i* that fall into class *j* during classification. The diagonal elements of the matrix reflect the number of correctly classified time series. The *OA* is the ratio of the sum of diagonal elements to the sum of all elements of the error matrix *N* (expressed as a percentage):(5)$$OA=\frac{\sum_{i=1}^{n}X_{ii}}{N}\;*\;100\%.$$

The confusion matrix presents both the number of time series in class *i* (*i* ∈ [1,*n*]) erroneously assigned to other classes (false negative time series, FN) and the number of pixels erroneously assigned to class *i* (false positive time series, FP) by the classifier. The F-measure for class *i* is a harmonic estimate of the FN and FP quantities:(6)$${F1}_{i}=\frac{2*\frac{{TP}_{i}}{{TP}_{i}+{FN}_{i}\;}\times \frac{{TP}_{i}}{{TP}_{i}+{FP}_{i}}}{\frac{{TP}_{i}}{{TP}_{i\;}+{FN}_{i}}+\frac{{TP}_{i}}{{TP}_{i}+{FP}_{i}}}.$$

To assess the model’s performance, OA and F1 metrics were calculated for different ML algorithms (RF, SVM, QDA). The accuracy metrics ${OA}_{test}$, ${F1}_{test}^{soy}$, ${F1}_{test}^{oat}$, ${F1}_{test}^{fallow}$ were evaluated. The best-performing method was selected based on the analysis of the accuracy metrics. This method was then used in the classification of land in the Amur Region. To assess the performance of the classifier in the neighboring region, the metrics ${OA}_{Amur}$, ${F1}_{Amur}^{soy}$, ${F1}_{Amur}^{oat}$, ${F1}_{Amur}^{fallow}$ were calculated.

To assess the efficiency of the classifier, it is also important to determine the number and proportion of fields in which crops were identified correctly. The number of time series assigned by the classifier to a particular class was counted to identify the crops in each field. Each field was assigned the label of the class with the greatest number of time series. We calculated the number and proportion of fields with correctly identified crops to assess classification accuracy at the field level.

## 3. Results and Discussion

### 3.1. DpRVI Seasonal Course

Figure 7 shows the seasonal course of DpRVI for soybean, oat, and fallow fields in the Khabarovsk District and the averaged course for the three districts of the Amur region that were studied. Table 5 shows the mean DpRVI maxima values (with variation among fields) and maxima dates (DOY) for the studied districts by crops.

Figure 7a,b show that soybean sowing in the Far East was carried out at the end of May and in the first half of June. However, it can be noted that in the Khabarovskiy District, a grass cover was formed on the fields by the beginning of June. In areas of the Amur Region, as a result of soil treatment with herbicides, the cover was clearly not formed. As soybean grows in the fields, the DpRVI value was also increased. The values of the DpRVI maximum varied insubstantially, from 0.58 for the Oktyabrskiy District to 0.61 for the Ivanovskiy District. The maximum was reached in the second half of September (DOY 259 (16 September) for the Khabarovskiy District and DOY 262–266 (19–23 September) for the districts of the Amur Region). From the end of September onwards, the DpRVI values began to decrease rapidly due to the wilting of plants. At this time, the soybean was ready for harvest, but the exact date of harvest often depends on weather conditions (in rainy seasons, harvest was only done at the beginning of winter).

Oat sowing was carried out in May; however, in the Khabarovskiy District, there was an early increase in DpRVI values before sowing the crop (Figure 7c,d). This growth can be caused not only by the rapid growth of weeds in May but also by incomplete harvesting in the previous agricultural season—the seeds left on the field yield early shoots. The DpRVI maximum for oat was reached in July and DpRVI reached a plateau during this period. Therefore, DOY of the maximum can vary significantly among fields—from the beginning of July in the Ivanovskiy District to the end of July in the Khabarovskiy District. The values of the maximum also varied considerably, from 0.55 in the Arkharinskiy District to 0.7 in the Ivanovskiy District. In the entire study area, oat harvesting was carried out in late July–August. Further growth of DpRVI in some fields may be caused by the growth of grasses after harvesting oat, which is especially true for fields where oats grow together with perennial grasses.

DpRVI course in fallow land did not show distinct peaks due to a lack of planting and harvesting during the entire study period (Figure 7e,f). Nevertheless, various natural and anthropogenic phenomena can lead to some anomalies in the growth of grass cover on abandoned cropland. Due to the heterogeneity of the fallows and the minor variation in the DpRVI during the season, the DpRVI maximum in different areas varied slightly (from 0.63 to 0.65); the maximum DOY can occur on any date during the season.

### 3.2. Results of Cropland Classification

#### 3.2.1. ML Hyperparameter Optimization

Based on the grid search results, RF hyperparameters were chosen during cross-validation. A Gini function was applied to measure the quality of a split. The minimum number of samples required to split an internal node was 2. The most important parameter influencing cross-validation accuracy was the number of trees. Figure 8 shows the cross-validation accuracy dynamics with the increasing number of trees. Accuracy increased slightly after 20 trees, peaked at 93.5% at 120 trees, and then overfitting was observed. Thus, 120 trees were chosen as the optimal value. Increasing this parameter further does not improve accuracy.

The SVM method also has a set of hyperparameters. A grid search revealed that the balanced class weight insubstantially increased the accuracy. Balanced weight uses the values of *y* to automatically adjust weights inversely proportional to class frequencies in the input data. L2 norm was used in the penalization, and squared hinge was the loss function. The maximum number of iterations was increased to 10,000. Figure 9 shows the tuning of regularization parameter C. It is a loss term that makes sure that that the weights classify the training data points correctly. The highest cross-validation accuracy (93.5%) was achieved with a standard value (C = 1), which means that changes in this hyperparameter decreased the accuracy.

#### 3.2.2. Khabarovsk District Classification Results

Table 6 presents the values of the accuracy metrics on the Khabarovsk District test set for different ML algorithms with optimized hyperparameters. The best performance was shown by the RF algorithm. RF ${OA}_{test}$ was 82.0%. ${F1}_{test}^{soy}$ exceeded 0.9, which indicates that the number of classifier errors was small. The value of ${F1}_{test}^{oat}$ and ${F1}_{test}^{fallow}$ was 0.78. From the point of view of the classifier, the oat class was most similar to fallow; 31% of oat time series were assigned to fallow. SVM metric values were similar to RF. ${OA}_{test}$ for the SVM with a linear kernel was 81.9%, QDA ${OA}_{test}$ was 81.1%, and the value of ${F1}_{test}^{oat}$ and ${F1}_{test}^{fallow}$ for QDA were 0.76 and 0.77, respectively.

Figure 10 shows the confusion matrix for the RF. The proportion of correctly identified time series was: 94% for fallow, 97% for soybean and 68% for oat. Further, 31% of oat time series were attributed by RF to the fallow class, and among the time series assigned by the classifier to the fallow class, the share of the oat was 27%.

We evaluated our classification results in comparison to the results of prior studies that used radar data. For example, the use of various combinations of polarizations and the RVI index by [22] when training the RF classifier made it possible to achieve an accuracy of 66%. In a study by [23], when classifying cropland in the Netherlands, OA was 69.5%. A combination of different polarizations and double polarization VIs was used as parameters for training by the nearest-neighbor method. Zhou et al. [47] achieved the highest overall classification accuracy (85%) using RF (VH-polarization); the accuracy of soybean determination was 88%.

Thus, the performance of our approach is quite high. For an additional assessment of the quality of the model, RF was applied on a sample from another region with similar climatic and soil conditions.

#### 3.2.3. Test Results in Amur Region

The RF classifier was applied to label the DpRVI time series of the fields in the test sample located in the Amur Region. The values of the metrics for assessing the efficiency of the classification of a test sample are presented in Table 7. When using the classifier in another region, ${OA}_{Amur}$ was 83.1%. F1 for soybean decreased by 0.06, but for fallow and oat it increased by 0.06 and 0.01, respectively.

Figure 11 shows the RF confusion matrix for the test set in Amur Region. The proportion of correctly classified time series was 87% for the fallow and soybean and 75% for oat. Due to regional differences in seasonal DpRVI profiles, 14% oat time series were identified as fallow. There were also some difficulties in the separation of soybean and oat—11% of the oat time series were recognized as soybean. In general, however, we can say that when using the classifier on the fields of another region, there was no decrease in accuracy.

### 3.3. Crop Identification at the Field Level

Table 8 presents the results of crop identification in the fields of the Amur Region. The overall identification accuracy was 79% (6 errors per 29 fields). All fields where soybean grew, as well as the fallow land, were recognized correctly.

Figure 12 shows examples of crop identification in the fields. Figure 12a shows two soybean and four abandoned fields, located in Arkharinskiy District. Out of 12,413 soybean field pixels, 10,148 (82%) were recognized correctly. Problematic areas in the fields were identified as a fallow. Similarly, 7916 out of 8497 fallow land pixels were identified correctly (93%). The detection of growing crops in individual pixels of agricultural fields may be associated with the germination of crops sown in previous years. Figure 12c shows the example of correct oat detection in two fields (total area was 463 ha) in Ivanovskiy District. RF detected 40,317 oat pixels (94% of total amount). Black sites (fallow land) indicated wetlands. Figure 12d shows an example of correct identification of abandoned land in Ivanovskiy District. RF identified 32,203 out of 36,362 time series (88.6%) as fallow land.

Meanwhile, only four of ten oat fields were correctly identified (examples of incorrect identification are presented in Figure 12b). Five incorrectly identified fields were recognized as soybean, one field was recognized as fallow land. Five of six fields were located in the Arkharinskiy District. Figure 13 shows the DpRVI seasonal course for oat in the training set (Khabarovskiy District), in the Ivanovskiy District (plus one field in the Oktyabrskiy District), where oat fields were recognized correctly, and in the Arkharinskiy District, where the classifier made a mistake. In the Khabarovskiy and Ivanovskiy districts, the course of DpRVI for most of the season had similar trends (peak in July and decrease in values after harvesting). Oat in the Arkharinskiy District was characterized by low DpRVI in summer and a sharp increase in autumn with the formation of a second peak, temporally coinciding with the maximum in soybean, which led to incorrect classification. The presence of the second peak indicated the growth of the second crop, which was not documented in the crop database. It is known that perennial grasses are sown with crops; sowing seeds of timothy grass, clover, and alfalfa with oat increases its productivity and therefore is a fairly common practice [48]. However, the presence of an additional crop that significantly affects the DpRVI seasonal course led to classification errors. In further studies, it is planned to separate the DpRVI time series of fields with overseeding of perennial grasses into a separate class.

Prior research has attempted to identify abandoned cropland. For example, Qiu et al. [14] used Sentinel-1 VV polarization time series and achieved an accuracy of 90% when classifying abandoned cropland at the field level in China. The use of the DpRVI time series in our work made it possible to achieve unmistakable classification (all 10 fields were identified correctly). The accuracy of crop identification in the fields has also been previously assessed using radar data [47]. In this study, fields with soybean were classified with an accuracy of 88% (the test sample consisted of 100 polygons). In general, even with a smaller number of fields and taking into account the transfer of learning outcomes to a neighboring region, the classification accuracy at the field level in our study is consistent with the findings of studies published in leading scientific journals.

### 3.4. Comparison with Another SAR Data Time Series

The use of radar data for cropland mapping is not new to scientific research. Among the most popular dual-pol Sentinel-1 data for the cropland mapping, one can single out the ratio of VV/VH polarizations and the RVI from Level-1 Ground Range Detected High Resolution (GRD-HR) product. GRD-HR consists of focused SAR data that have been multi-looked and projected to ground range using an Earth ellipsoid model of resolution 10 m × 10 m on pixel.

We applied the approach proposed for DpRVI to the VV/VH polarization ratio and the RVI time series at the same sites. The number of time series of radar data used for training and testing DpRVI-, VV/VH-, and RVI-based models in the Khabarovskiy District, as well as number of time series in the Amur region, are presented in Table 9. As already mentioned in Section 2.3, we used 71,153 DpRVI time series to train the classifier, 30,550 series were used to test the Khabarovsk region, and 203,120 series were used to test the model on the fields of the Amur region. The number of time series obtained for the studied fields from the same images is higher for RVI and VV/VH than for DpRVI by 8–9%. RVI seasonal courses for the studied crops and study areas, as well as oat RVI courses in different districts are presented in Appendix A.

We applied RF to train classifiers based on RVI and VV/VH time series. Table 10 and Table 11 show that RVI and VV/VH classifiers demonstrated similar results and were less accurate in comparison with the DpRVI model. For the Khabarovskiy District, RVI ${OA}_{test}$ was 77% (the difference with DpRVI was 5%) and VV/VH ${OA}_{test}$ was 76.8% (5.2% less than DpRVI). F1 for RVI and VV/VH varied from 0.73 for the fallow class to 0.85 for the soybean class (0.04–0.07 less than DpRVI). RVI and VV/VH model application in the Amur region led to a decrease in classification accuracy by 2%. At the same time, just as for DpRVI, F1 for soybean decreased by 0.06, whereas oat and fallow F1 remained at the same level.

Thus, we have shown that the use of the DpRVI series makes it possible to increase the accuracy of classification, in comparison with the commonly used radar indicators for cropland mapping. These results can be related to the fact that DpRVI is calculated using the SLC product, which contains both amplitude and phase information from which we can generate a covariance matrix and obtain many polarimetric features. This makes the DpRVI index a more interpretable and sustainable descriptor of the crop growth course. It is also worth noting that DpRVI effectively incorporates scattered wave information to describe phenological changes which are vital for monitoring crop time series. In turn, data processing for DpRVI is more complex and requires high-performance computing.

## 4. Conclusions

This study successfully established the possibility of using DpRVI time series for cropland classification in Khabarovsk District. This process leads to good results in the identification of three classes: soybean, fallow, and oat. The seasonal course of DpRVI for each class follows a similar pattern, with similar maxima and the same calendar day on which these peaks are reached. According to the results of classification using three machine learning methods, it was found that the overall accuracy of the RF, SVM, and QDA methods was 82.0%, 81.9%, and 81.1%, respectively. The RF classifier was applied for cropland mapping in another region (Amur region). The use of RF for the fields of the Amur region did not worsen the performance; the overall accuracy was 83.1%, and F1 was 0.86 for soybean, 0.84 for fallow, and 0.79 for oat. The accuracy of crop identification at the field level was 79%. In future studies, it is recommended to separate oats that have been over-seeded with perennial grasses into a separate class to improve the accuracy of identifying the oat class. The proposed method aims to address the issue of classifiers, which rely on optical data, being affected by cloudiness that is common in the Russian Far East. For the Heilongjiang Province (northeastern China) cropland, the main phases of crop development, including soybean, coincided on a time scale with the phases of crop development in the regions of the Far Eastern Federal District, and the seasonal course of vegetation indices are quite similar [42,49]. Thus, the proposed approach to approximation of time series and cropland classification can be applied in regions with climatic conditions corresponding to the conditions of the southern part of the Khabarovsk Territory. It will also be able to classify fields in municipalities for which there is no ground reference data.

## Figures and Tables

**Figure 1 sensors-23-07902-f001:**
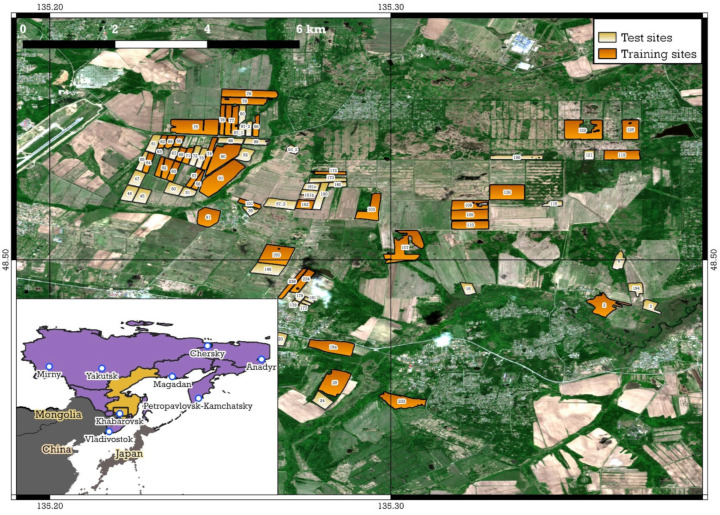
Study area 1 (Khabarovskiy District with internal numbers of sites).

**Figure 2 sensors-23-07902-f002:**
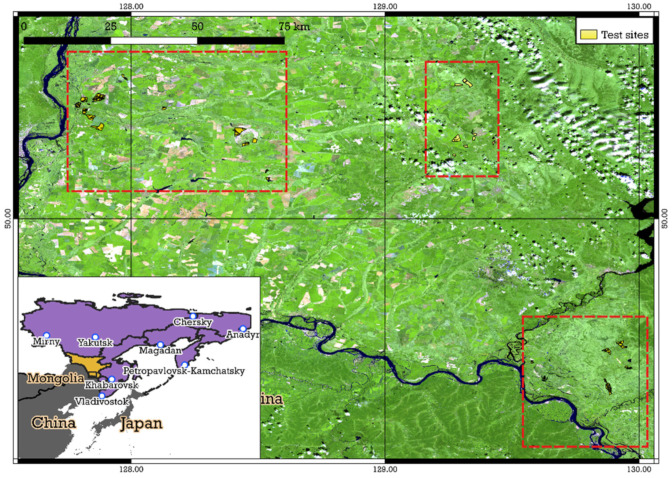
Study area 2 (Amur Region). Red boxes means spatial areas with sites.

**Figure 3 sensors-23-07902-f003:**
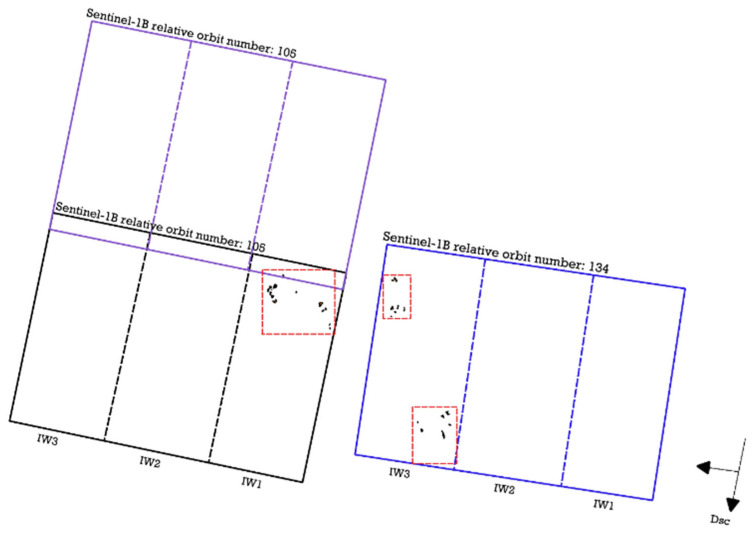
Sentinel-1 tracks for SA2. Red boxes means spatial areas with sites.

**Figure 4 sensors-23-07902-f004:**
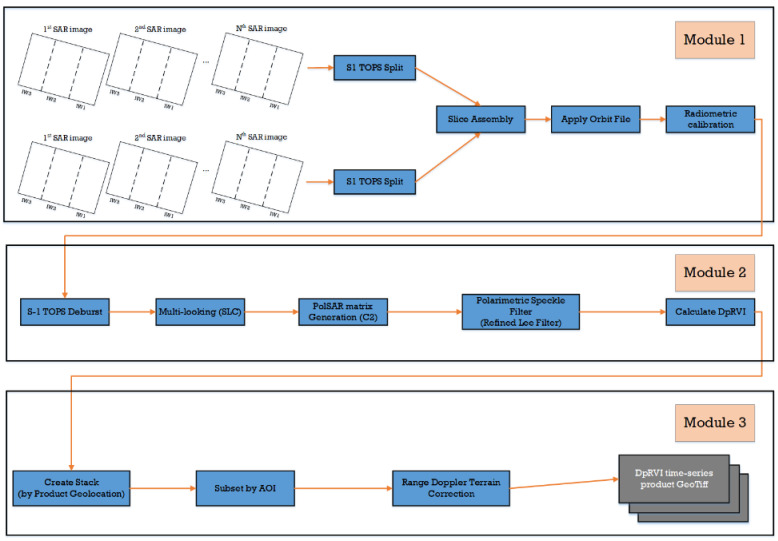
Flowchart of DpRVI time series generation for assembly of two slices.

**Figure 5 sensors-23-07902-f005:**
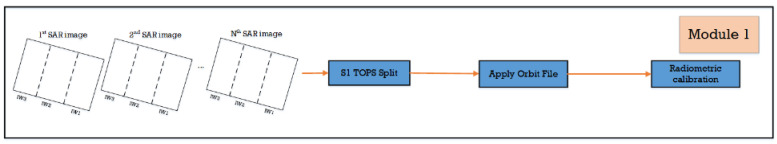
Flowchart for single sub-swath processing.

**Figure 6 sensors-23-07902-f006:**
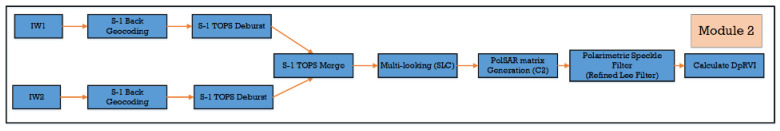
Flowchart for multiple sub-swath processing.

**Figure 7 sensors-23-07902-f007:**
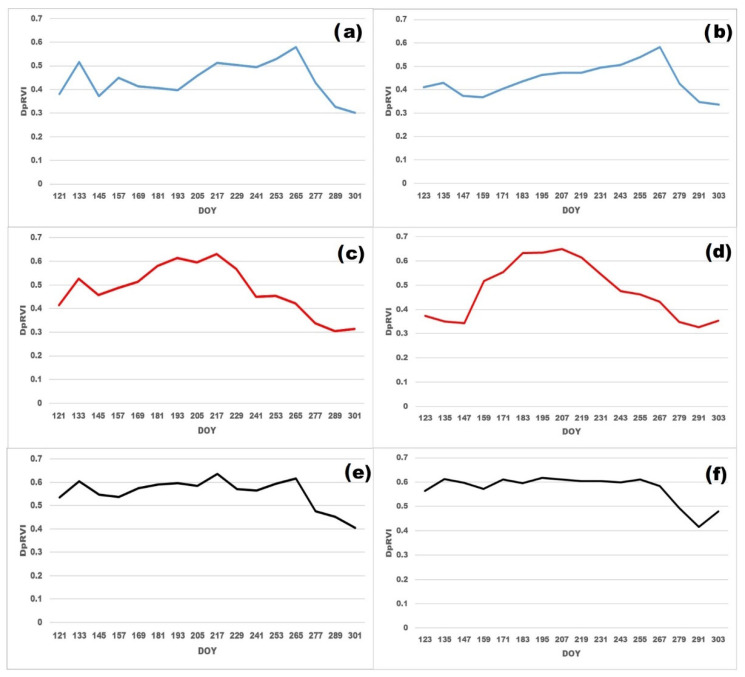
DpRVI seasonal course for: (**a**) soybean in Khabarovskiy District, (**b**) soybean in Amur Region, (**c**) oat in Khabarovskiy District, (**d**) oat in Amur Region, (**e**) fallow in Khabarovskiy District, (**f**) fallow in Amur Region.

**Figure 8 sensors-23-07902-f008:**
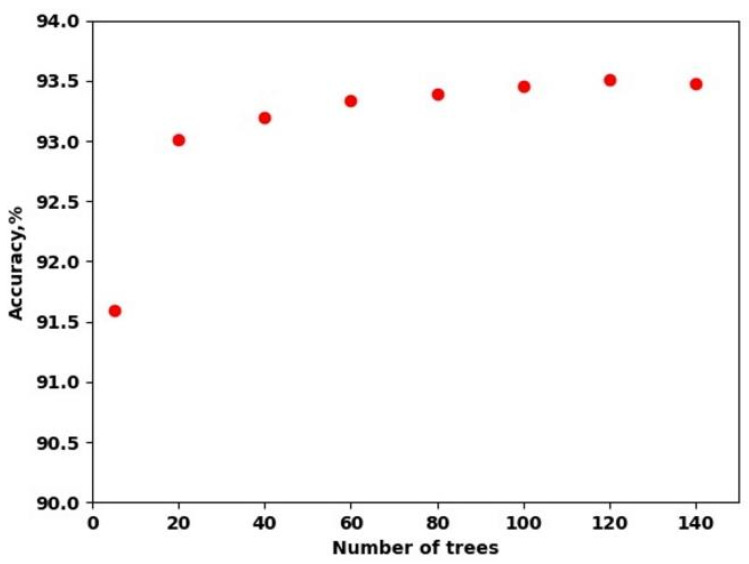
Trend of the cross-validation accuracy with the increasing number of random forest trees.

**Figure 9 sensors-23-07902-f009:**
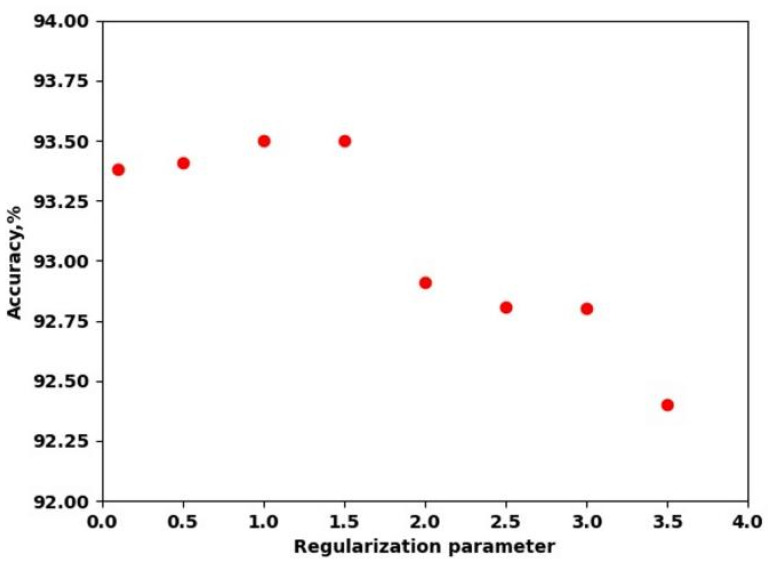
Cross-validation accuracy with the regularization parameter value.

**Figure 10 sensors-23-07902-f010:**
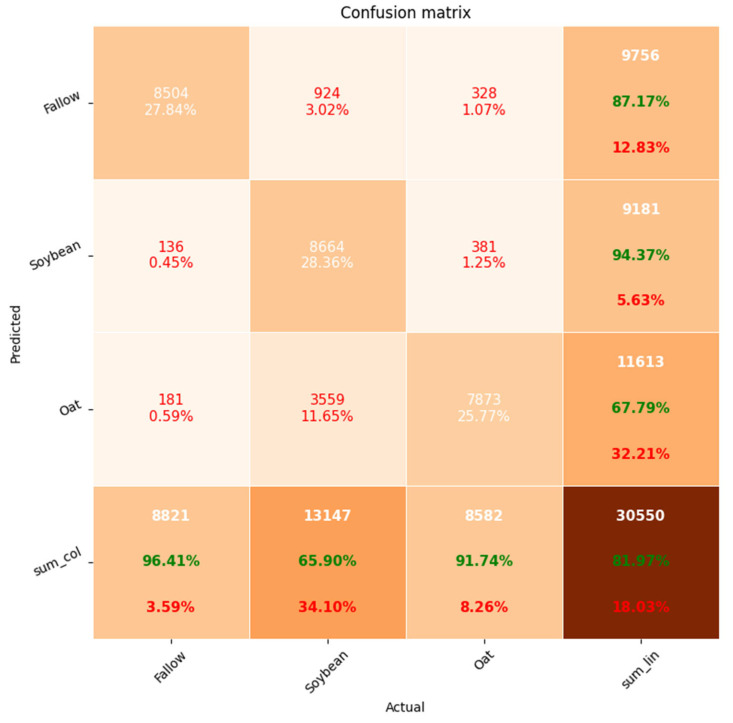
RF confusion matrix (Khabarovsk District).

**Figure 11 sensors-23-07902-f011:**
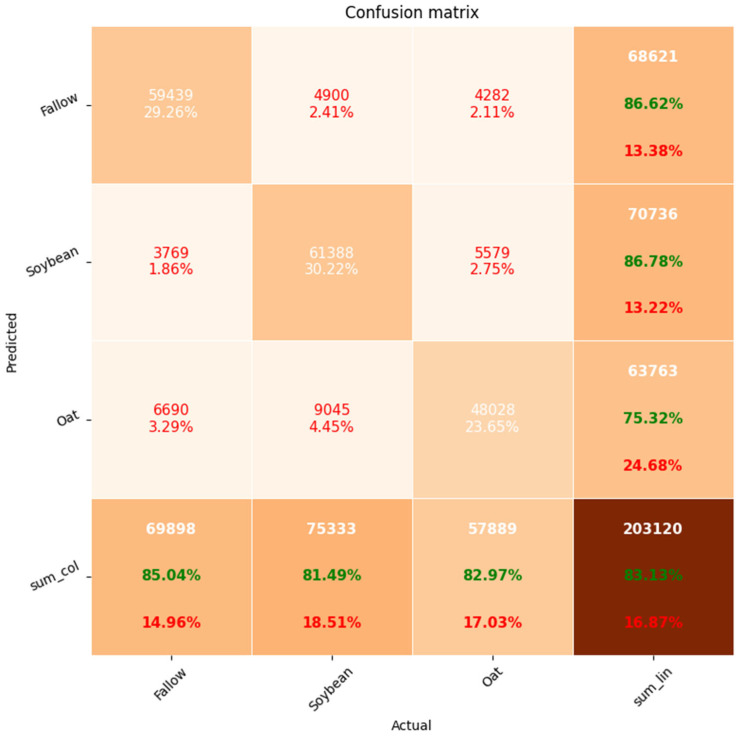
RF confusion matrix (Amur Region).

**Figure 12 sensors-23-07902-f012:**
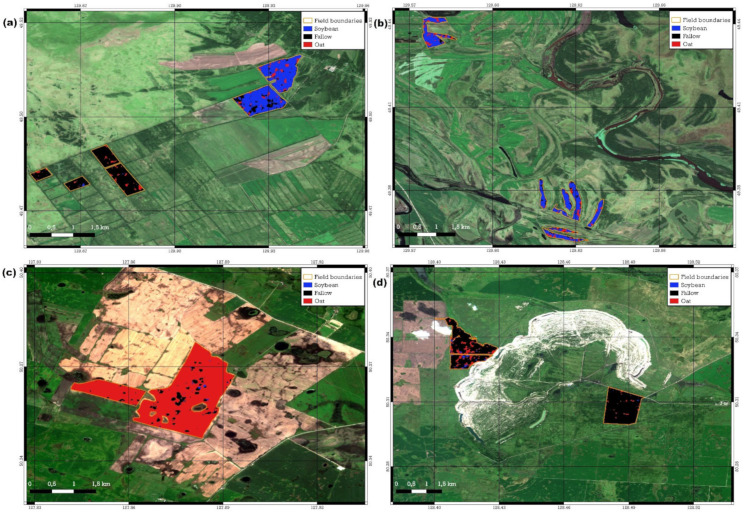
Examples of crop identification.

**Figure 13 sensors-23-07902-f013:**
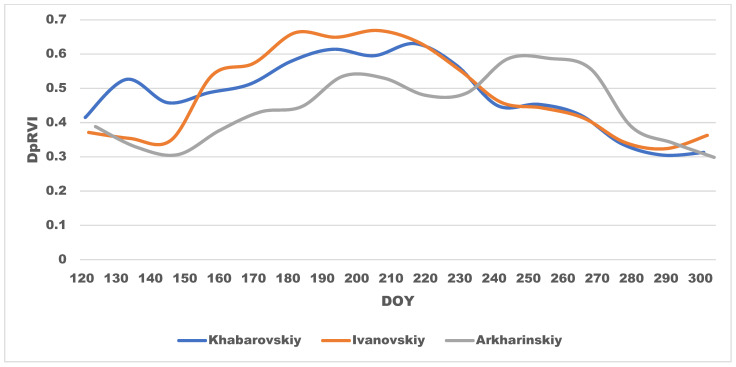
Oat DpRVI seasonal courses in the study districts.

**Table 1 sensors-23-07902-t001:** Specification for SAR images by Sentinel-1 data acquired for thestudy areas.

Study Area	Observation Period	Interferometric Wide Swath	Incidence Angle Range (Deg.)	Orbit	Relative Orbit Number	Azimuth and Range Resolution	Polarization
SA1	1 May 2021–28 October 2021	IW3	38.3–43.1	Descending	90	5 m × 20 m	VV, VH
SA2	2 May 2021–29 October 2021 (except 18 August 2021)	IW1	30.22–32.47	Descending	105	5 m × 20 m	VV, VH
	4 May 2021–31 October 2021 (except 3 July 2021)	IW3	38.3–43.1	Descending	134	5 m × 20 m	VV, VH

**Table 2 sensors-23-07902-t002:** Number of fields and their areas, included in training dataset.

Class	Training Dataset		Test Dataset	
	Fields	Area, ha	Fields	Area, ha
Fallow	10	299	10	109
Soybean	10	292	10	115
Oat	20	257	20	140
Overall	40	848	40	364

**Table 3 sensors-23-07902-t003:** Number of fields and their areas included in the Amur region dataset.

Class	Fields	Area, ha
Fallow	10	769
Soybean	9	775
Oat	10	780
Overall	29	2324

**Table 4 sensors-23-07902-t004:** Number of training and test DpRVI time series.

Class	Training	Validation	Amur Testing
Fallow	20,442	9181	70,736
Soybean	25,436	9756	68,621
Oat	25,275	11,613	63,763
Overall	71,153	30,550	203,120

**Table 5 sensors-23-07902-t005:** Mean DpRVI maxima values (with variation among fields) and maxima dates (DOY) for the studied districts by crops.

Crop	Khabarovskiy		Arkharinskiy		Ivanovskiy		Oktyabrskiy	
	Max	DOY	Max	DOY	Max	DOY	Max	DOY
Soybean	0.59 ± 0.01	258.9 ± 4.2	0.59 ± 0.01	264.0 ± 3.5	0.61 ± 0.01	266.0 ± 0.0	0.58 ± 0.04	261.75 ± 5.6
Oat	0.66 ± 0.01	212.5 ± 3.6	0.55 ± 0.02	202.0 ± 6.9	0.70 ± 0.02	190.0 ± 34.4	0.66 *	206 *
Fallow	0.65 ± 0.01	206.2 ± 11.5	0.64 ± 0.03	185.3 ± 40.3	0.63 ± 0.03	215.0 ± 90.1	0.65 ± 0.02	147.3 ± 34.1

* There was only one oat field in Oktyabrskiy District.

**Table 6 sensors-23-07902-t006:** Test accuracy for different ML methods in Khabarovskiy District.

Metric	RF	SVM	QDA
${OA}_{test}$,%	82.0	81.9	81.1
${F1}_{test}^{soy}$	0.92	0.92	0.92
${F1}_{test}^{oat}$	0.78	0.77	0.76
${F1}_{test}^{fallow}$	0.78	0.78	0.77

**Table 7 sensors-23-07902-t007:** RF test accuracy in Amur Region.

Metric	Value
${OA}_{Amur}$,%	83.1
${F1}_{Amur}^{soy}$	0.86
${F1}_{Amur}^{oat}$	0.79
${F1}_{Amur}^{fallow}$	0.84

**Table 8 sensors-23-07902-t008:** Crop identification accuracy in Amur Region.

Crop	Correctly Classified	Number of Fields	Accuracy, %
Soybean	9	9	100
Oat	4	10	40
Fallow	10	10	100
Overall	23	29	79

**Table 9 sensors-23-07902-t009:** Number of SAR time series.

SAR Data	Class	Training	Khabarovskiy Testing	Amur Testing
DpRVI	Fallow	20,442	9181	70,736
	Soybean	25,436	9756	68,621
	Oat	25,275	11,613	63,763
	Overall	71,153	30,550	203,120
VV/VH	Fallow	22,306	10,024	75,272
	Soybean	27,734	10,651	74,001
	Oat	27,582	12,659	70,327
	Overall	77,622	33,334	219,600
RVI	Fallow	22,306	10,024	75,262
	Soybean	27,734	10,651	74,024
	Oat	27,583	12,659	70,327
	Overall	77,623	33,334	219,613

**Table 10 sensors-23-07902-t010:** RF test accuracy for different SAR time series in Khabarovskiy District.

Metric	DpRVI	RVI	VV/VH
${OA}_{test}$,%	82.0	77.0	76.8
${F1}_{test}^{soy}$	0.92	0.85	0.85
${F1}_{test}^{oat}$	0.78	0.74	0.73
${F1}_{test}^{fallow}$	0.78	0.73	0.73

**Table 11 sensors-23-07902-t011:** RF test accuracy for different SAR time series in Amur region.

Metric	RF	RVI	VV/VH
${OA}_{Amur}$,%	83.1	74.9	74.6
${F1}_{Amur}^{soy}$	0.86	0.79	0.79
${F1}_{Amur}^{oat}$	0.79	0.72	0.72
${F1}_{Amur}^{fallow}$	0.84	0.73	0.73

## Data Availability

Data is contained within the article or Appendix A.

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
