# Peer review of "Cropland Mapping Using Sentinel-1 Data in the Southern Part of the Russian Far East"

_sensors, 2023, doi:10.3390/s23187902_

Round 1
Reviewer 1 Report (New Reviewer)
Authors presented good results on the use of radar satellite data in crop mapping. The study design is appropriate and the methods used are widely known in the crop remote sensing community. The paper will be useful to readers. However, authors should look at the following issues in the revision.
1) The introduction section is too long. I think most of the studies reviewed in the introduction can be more useful in the results and discussion sections.
2) What are the reasons for selecting SLC over GRD images in this study?
3) The conclusion seems more of a summary of findings rather than making inferences based on methods used and results obtained. Thus it should be rewritten.
The paper is generally well written. Only careful proofreading is needed.
Author Response
Thank you for your valuable comments.
Comment 1: The introduction section is too long. I think most of the studies reviewed in the introduction can be more useful in the results and discussion sections.
Response 1: Introduction has been shortened. Reference list has been updated.
Comment 2: What are the reasons for selecting SLC over GRD images in this study?
Response 2: The calculation of the DpRVI index occurs only using the SLC. For RVI and VH/VV the GRD level is used. Therefore, here in the text of the manuscript we note that in classification problems, the use of DpRVI makes it possible to increase its accuracy (due to m and Beta in formula 4 by SLC).
Comment 3: The conclusion seems more of a summary of findings rather than making inferences based on methods used and results obtained. Thus, it should be rewritten.
Response 3: The conclusion has been expanded. We have added information about the possibility of applying our results outside the study region.

Reviewer 2 Report (New Reviewer)
The manuscript deals with the common and relevant problem of crop identification. The authors present a workflow based on radar data to monitor vegetation growth and classify crops. The workflow contains three main steps that by themselves are not completely new, but their combination and the geographic area. The description of the methodology, results, discussion, and conclusions is easy to follow and thorough. The introduction is mainly focused on the technical parts (radar processing and machine learning algorithms). These parts are very detailed, but the justification of the research is less extensively covered. I suggest improving this section (lines 32-35, 158-180) and explaining why it is also relevant for other areas outside Russia.
Some more detailed remarks:
- It would be good to extend the titles of figure 1 and 2 with the name of the study areas (Khaburovskiy District and Amur Region) because in the text references are made to their names
- Figure 1 and 2: it is difficult to understand the geography of the maps. It would be good to add other county boundaries and names to the overview map. For figure 1 another inset map that shows the district, with some city names would greatly help.
- line 303, and figure 1. It would be good to show the training and test sites with different colors
- Use of DOY in Figure 7, 13 and in the text: It would be good to help the reader by adding real dates to the figures and text, because now the reader needs to convert the DOYs to dates himself. In the figures they could be show both, DOY and dates
- Figure 10 and 11, it would be good to add the relative share of the total number of pixels in each cell, e.g in brackets below the absolute number.
- Line 535. It seems that visually the pixel-based classification results were converted fields. Is this correct? Is there maybe a method that is less arbitrary? Were there any fields that showed a lot of mixed pixels?
- Section 3.4 The authors calculated RVI as well. It would be interesting to create an RVI time series as well and compare it with the DpRVI time series (as in figures 7 and 13).
- Line 610. The authors show that the accuracy is improved using DpRVI. In this discussion, I miss that complexity of the workflow, which is much higher for DpRVI, due to the use of SLC data compared to the RVI which can be calculated based on GRD data.
There are only very few minor errors in the English
Author Response
Thank you for your important remarks. We believe that corrections made to the text allowed to improve quality of the manuscript.
Comment 1: I suggest improving this section (lines 32-35, 158-180) and explaining why it is also relevant for other areas outside Russia.
Response 1: Some information about possibility to use results of our research in other areas (on China example) has been added to Conclusions.
Comment 2: It would be good to extend the titles of figure 1 and 2 with the name of the study areas (Khabarovskiy District and Amur Region) because in the text references are made to their names.
Response 2: Titles have been extended.
Comment 3: Figure 1 and 2: it is difficult to understand the geography of the maps. It would be good to add other county boundaries and names to the overview map. For figure 1 another inset map that shows the district, with some city names would greatly help.
Response 3: Additional information about neighboring countries and cities has been added. Image has been updated.
Comment 4: Line 303, and figure 1. It would be good to show the training and test sites with different colors.
Response 4: Training and test sites have been highlighted in Figure 1. Test sites have been highlighted in Figure 2.
Comment 5: Use of DOY in Figure 7, 13 and in the text: It would be good to help the reader by adding real dates to the figures and text, because now the reader needs to convert the DOYs to dates himself. In the figures they could be show both, DOY and dates.
Response 5: We suppose that specifying the DOY and the date both is unnecessary and will make the images overloaded. At the same time, we have indicated the date in parentheses in all references to DOY in the text.
Comment 6: Figure 10 and 11, it would be good to add the relative share of the total number of pixels in each cell, e.g in brackets below the absolute number.
Response 6: Confusion matrices were generated using another plot library. Now, they contain both the number of pixels and the relative shares.
Comment 7: Line 535. It seems that visually the pixel-based classification results were converted fields. Is this correct? Is there may be a method that is less arbitrary? Were there any fields that showed a lot of mixed pixels?
Response 7: Pixel-based classification was carried out separately, and field classification is an aggregation of pixel classification. So obviously there were fields with mixed results. Examples of fields with a sufficiently large number of mixed pixels are presented, for example, in Figures 12a and 12c; Figure 12b shows the result of incorrect classification. The practically important assessment of field-level accuracy is scientifically less arbitrary. That is why our work provides accuracy at both the pixel level and the field level.
Comment 8: Section 3.4 The authors calculated RVI as well. It would be interesting to create an RVI time series as well and compare it with the DpRVI time series (as in figures 7 and 13).
Response 8: RVI seasonal course for studied crops and study areas, as well as oat RVI courses in different districts have been presented in Figure S1 and Figure S2 in Supplementary Files.
Comment 9: Line 610. The authors show that the accuracy is improved using DpRVI. In this discussion, I miss that complexity of the workflow, which is much higher for DpRVI, due to the use of SLC data compared to the RVI which can be calculated based on GRD data.
Response 9:
We have shown that the use of the DpRVI series makes it possible to increase the accuracy of classification, in comparison with the commonly used radar indicators for cropland mapping. It is also worth noting that DpRVI effectively incorporates scattered wave information to describe phenological changes which are vital for monitoring crop time series. The last paragraph in the discussion has been changed. We believe that the changes made will improve the article (Lines 624-626).

This manuscript is a resubmission of an earlier submission. The following is a list of the peer review reports and author responses from that submission.
Round 1
Reviewer 1 Report
The present study tried to map the cropland using Sentinel-1 data in Russia. My major comments are
I strongly suggest the authors to optimise the hyperparameters of machine learning models using cross validation or any other bio-inspired models.
Please write the results in past tense
L493-494: The accuracy of fallow and oat classification was higher during validation then training. What may be the reason(s) for that?
What was the package used for raster data handling in Python?
What are the limitations of the present study?
DpRVI was calculated using which software? Please mention that.
Some specific comments are:
L12: “abandoned cropland” to “fallow land”
L22: What about the results of SVM and RF? Whether QDA was the best performing model which I could find in results and discussion section. Please mention in abstract that QDA was the best performing model.
L26: Delete “with ground reference data absence”
L39: “optical vegetation indices (VI) time series” to “time series of optical vegetation indices (VI)”
L45-46: “SVM and RF methods were used for classification; the highest…” to “The highest…”
L49-50: “SVM, its modification Gray Wolf Optimizer support vector machine (GWO-SVM)” to “Gray Wolf Optimizer with support vector machine (GWO-SVM)”
L57-58: “Early classification was also studied - the possibility” to “The possibility”
L84: “temporary VV polarization time series” to “time series of VV polarization”
L86: “scientists use either” to “scientists have used either”
L92: “particular crop can help perform” to “particular crop can help to perform”
L93-34: “however, it does not allow the evaluation of its spatial distribution and build crop maps.” How? Can you elaborate?
L104: Rewrite “Multiyear studies allow you to control crop rotation.” to make it clear.
L140-141: Delete “One of the limitations of crop mapping in the single region is the possibility of using the model only in one region.”
L143-144: “Russia, have been faced with the problem” to “Russia, have faced the problem”
L145: “temporary” to “temporal”
L158: “performance” to “performed”
L174: “them According” to “them. According”
L208: Rewrite the sentence
L285: Mention the UTM zone number
L302: What do you mean by “expert analysis”? Visual interpretation?
L336: “performed: for each field” to “performed. For each field”
L341: “which consisted of 262,678 pixels” for one image?
Table 4: What does these values indicates? Please elaborate. The title of tables and figures should be standalone.
L360: Whether “LinearSVC” was used in this study? Please write it clearly.
L365: Final output is the average of all these trees for regression problem while it is majority votes for classification problem. So, correct the statement.
L370: “the number of trees in our model to 5” The number is very small which may lead to under fitting. We generally use 500 trees.
L379-380: Please rewrite the sentence
Author Response
Comment 1: Extensive editing of English language and style required.
Response 1: We used the service "Cambridge Proofreading" for grammar, punctuation and style editing.
Comment 2: I strongly suggest the authors to optimise the hyperparameters of machine learning models using cross validation or any other bio-inspired models.
Response 2: GridSearchCV method (scikit-learn) have been applied to optimize SVM and RF hyperparameters. 10-fold cross-validation have been performed for hyperparameters tuning. Corresponding changes have been represented in Sections 2.4, 2.5, 3.2.
Comment 3: Please write the results in past tense.
Response 3: Results were written in past tense.
Comment 4: L493-494: The accuracy of fallow and oat classification was higher during validation then training. What may be the reason(s) for that?
Response 4: It was validation peculiarities. We have performed accurate 10-fold cross validation and recalculate results of cross-validation. has been 0.86, has been 0.76. has been 0.94, has been 0.88. Remind: these are changes in the cross-validation calculating. It does not affect the results on the test set.
Comment 5: What was the package used for raster data handling in Python?
Response 5: Processing of raster was carried out using the python modules as GDAL and RasterIO.
Comment 6: What are the limitations of the present study?
Response 6: Main limitation is a small amount of reliable ground reference data. This is what limited our study area. Another problem is cropland homogeneity due soybean domination. Regarding the limitations in using the proposed technique, the main is interregional difference in DpRVI seasonal course. That’s why we tested our model in neighbor region with the similar climatic conditions. All of the above is described in the text of the article.
Comment 7: DpRVI was calculated using which software? Please mention that.
Response 7: We clearly indicate standard processing steps for the DpRVI [42] calculations. We used Graph Processing Framework in SNAP.
Comment 8: L12: “abandoned cropland” to “fallow land”.
Response 8: Corrected.
Comment 9: L22: What about the results of SVM and RF? Whether QDA was the best performing model which I could find in results and discussion section. Please mention in abstract that QDA was the best performing model.
Response 9: SVM and RF cross-validation results and reasons to choose QDA method is presented in Section 3.2.1. Information that QDA is the best model have been added to abstract.
Comment 10: L26: Delete “with ground reference data absence”.
Comment 11: L39: “optical vegetation indices (VI) time series” to “time series of optical vegetation indices (VI)”
Comment 12: L45-46: “SVM and RF methods were used for classification; the highest…” to “The highest…”
Comment 13: L49-50: “SVM, its modification Gray Wolf Optimizer support vector machine (GWO-SVM)” to “Gray Wolf Optimizer with support vector machine (GWO-SVM)”.
Comment 14: L57-58: “Early classification was also studied - the possibility” to “The possibility”
Comment 15: L84: “temporary VV polarization time series” to “time series of VV polarization”.
Comment 16: L86: “scientists use either” to “scientists have used either”
Comment 17: L92: “particular crop can help perform” to “particular crop can help to perform”
Response to comments 10-17: Thanks for language editing. All comments have been taken into account, and the wording has been corrected.
Comment 18: L93-94: “however, it does not allow the evaluation of its spatial distribution and build crop maps.” How? Can you elaborate?
Response 18: Sentence was deleted from manuscript. Work 17 was re-examined. Indeed, the proposed method makes it possible to build cropland maps.
Comment 19: L104: Rewrite “Multiyear studies allow you to control crop rotation.” to make it clear.
Response 19: Changed to “Time series for several years make it possible to observe crop rotation”.
Comment 20: L140-141: Delete “One of the limitations of crop mapping in the single region is the possibility of using the model only in one region.”
Response 20: Sentence was deleted.
Comment 21: L143-144: “Russia, have been faced with the problem” to “Russia, have faced the problem”
Comment 22: L145: “temporary” to “temporal”
Response 21-22: Corrected.
Comment 23: L158: “performance” to “performed”
Response 23: Rephrased to “was well performed”
Comment 24: L174: “them According” to “them. According”
Response 24: Sorry, corrected.
Comment 25: L208: Rewrite the sentence
Response 25: Rewrited to simply “Study area 2”
Comment 26: L285: Mention the UTM zone number
Response 26: SAR stack data were geocoded to a WGS84 / UTM zone 52N projected coordinate system. Information have been added to manuscript.
Comment 27: L302: What do you mean by “expert analysis”? Visual interpretation?
Response 27: Yes, you are right. It is visual analysis of crop growth dynamics in the studied sites using satellite images.
Comment 28: L336: “performed: for each field” to “performed. For each field”
Response 28: Corrected.
Comment 29: L341: “which consisted of 262,678 pixels” for one image?
Response 29: We classified DpRVI time series. Each series contains 16 observations. 262,678 pixels from each image was utilized to build series set. So, “pixels” have been rephrased to “time series” in manuscript.
Comment 30: Table 4: What does these values indicates? Please elaborate. The title of tables and figures should be standalone.
Response 30: These values indicate number of time series in training and validation datasets and proportions of time series between classes. Table caption was changed to “Training and validation time series”.
Comment 31: L360: Whether “LinearSVC” was used in this study? Please write it clearly.
Response 31: Linear Support Vector Machine is scikit-learn implementation of Support Vector Machine with linear kernel. Hyperparameters of Linear SVC have been added to manuscript.
Comment 32: L365: Final output is the average of all these trees for regression problem while it is majority votes for classification problem. So, correct the statement.
Response 32. You are right. It was implementation of RF for regression task. The statement has been corrected.
Comment 33: L370: “the number of trees in our model to 5” The number is very small which may lead to under fitting. We generally use 500 trees.
Response 33: We performed hyperparameters search using GridSearch. Optimal number of trees is 120. We have added a graph of learning accuracy versus the number of trees. You can see that, in general, after 60 trees accuracy increased insubstantially and after 120 trees overfitting was observed. 5 trees are really too few, but 500 trees are redundantly.
Comment 34: L379-380: Please rewrite the sentence.
Response 34: We decided to delete this sentence, because same idea about QDA was presented few sentences later.

Reviewer 2 Report
The article looks like a work report, the scientific nature of the article must be improved. The introduction is very wordy and needs to be further condensed. In the discussion section, the dialogue with the existing literature should be enhanced.
Author Response
Comment: The article looks like a work report, the scientific nature of the article must be improved. The introduction is very wordy and needs to be further condensed. In the discussion section, the dialogue with the existing literature should be enhanced.
Response: Due reviewers comments, we have changed cross-validation calculation (from 3-fold to 10-fold), optimized machine learning hyperparameters. Hyperparameters optimization has increased Random Forest and Support Vector Machines performance. The results of experiments have been visualized in main parameters vs accuracy graphs. Other parameters have been presented in manuscript text. Some technical information about image processing have been added. These changes and changes related with minor reviewers’ comments have improved scientific nature of manuscript. We suppose that introduction fully represents the current state of cropland mapping using remote sensing. Relevant works was presented both for the use of optical and radar images, the advantages and disadvantages of studies was described. In the discussion, we compared our results with the works closest to our study.

Reviewer 3 Report
The paper addresses the application of supervised machine learning classifications between two crops and fallows using SAR image to deal with the weather limitation of optical images as well as the issues of using multi-temporal data.
The methodology and the results are well presented and the paper is well written. However, this topic has been well-explored and the current study is a replication of what have already been done in existing literature.
Also several authors have used S-1 dual polarization information and classifiers for crop and land cover mapping with better results than the present study.
I cannot recommend this manuscript for publication due to its lack of novelty. In current literature, the authors failed to review the use of single optical image for crop type mapping in introduction, which is a good solution to cloudy area and for near-real time mapping. The study was awkwardly designed to compare 3 machine learning classifications and strangely RF does not give good results unlike the majority of existing works that prove its stable performance and less affected by overfitting. The parameters used in RF as a number of 5 trees is not sufficient as a general rule of thumb, values ranging from 50 to 400 trees are a fairly reliable range of values that will cover most cases.
It is true that DpRVi is widely used for crop growth monitoring and few works have tested it for crop discrimination, but the classification method presented here only discriminates between two crops and fallow land. This means that we must first have the areas cultivated or occupied by these three classes in order to make a land use map. What about other major crops or land cover classes such as grasses, shrubs, bare land, built-up, waterbody. Hence I don't see the usefulness of the results of the article for managers, for example the authors mention that the managers will use the method to detect abandoned land and fallow land, whereas the method itself requires to know first where the fallow land and the soybean and oat fields are in order to process.
Author Response
Comment 1: This topic has been well-explored and the current study is a replication of what have already been done in existing literature. Also, several authors have used S-1 dual polarization information and classifiers for crop and land cover mapping with better results than the present study. It is true that DpRVI is widely used for crop growth monitoring and few works have tested it for crop discrimination.
Response 1: Yes, many authors have used SAR data for crop mapping. Usually, they utilized polarizations (VV, VH, VH/VV ratio). Several authors classified cropland using radar vegetation indices (RVI, DpSVI). The novelty of this study is the DpRVI application in crop mapping. There are several works dedicated to DpRVI. For example, Mandal et al. [29] declares that DpRVI is sensitive to crop growth and is used as a relatively simple and physically interpretable vegetation descriptor. Also, Mandal et al. shows the curves of the seasonal course for different crops. Later, DpRVI can be evaluated through comparison against NDVI: these indices have high correlation and determination. These results lead to the possibility of using DpRVI for the crop mapping. Such cropland classification is carried out in our study and, thus, it is a logical development of scientific thought.
Comment 2: In current literature, the authors failed to review the use of single optical image for crop type mapping in introduction, which is a good solution to cloudy area and for near-real time mapping.
Response 2: We supposed that is not the best idea, especially for several geographically distributed study areas. The one image approach requires the absence of cloudiness in a certain period of time in several municipalities at once, which is a rather rare event in summer. In addition, we are doing research on the use of optical indexes for cropland mapping and are aware of the difficulties associated with its use in our region, that is very cloudy in summer. Studying the literature on cropland classification based on optical vegetation indices, we noticed that the development of this area of knowledge is moving away from classification by one image to the use of time series and composites. In the case of classification by landcover types, the single-image approach can give good results. in the case of crop mapping, it is not very promising due to the intersection of NDVI plots for different crops - only seasonal dynamics can give reliable information. We have cited a number of references to studies that show the advantage of using time series, including for optical indices [4-6].
Comment 3: The study was awkwardly designed to compare 3 machine learning classifications and strangely RF does not give good results unlike the majority of existing works that prove its stable performance and less affected by overfitting.
Response 3: RF is the most common algorithm. But it doesn’t guarantee its superiority over other methods. For example, in [45-47] SVM was used. Zhang, et al. [8] proposed that modification of SVM allows to build more precised map than RF. Our experiments revealed that in our case RF have high, but not the highest accuracy.
Comment 4: The parameters used in RF as a number of 5 trees is not sufficient as a general rule of thumb, values ranging from 50 to 400 trees are a fairly reliable range of values that will cover most cases.
Response 4: Yes, you are right. We have performed additional hyperparameter tuning. The overall accuracy peaks at 93.5% at 120 trees, and then overfitting was observed. Thus, 120 trees were chosen as the optimal value. Information about hyperparameters have been added to manuscript.
Comment 5: What about other major crops or land cover classes such as grasses, shrubs, bare land, built-up, waterbody. Hence, I don't see the usefulness of the results of the article for managers, for example the authors mention that the managers will use the method to detect abandoned land and fallow land, whereas the method itself requires to know first where the fallow land and the soybean and oat fields are in order to process.
Response 5: Cropland area detection was not our task in current research. Several algorithms for land use mapping were presented in scientific papers (including use of single optical images). For example, Bartalev, et al. [2] built land cover maps in all-Russian and global scale. In our case, we worked with ready field shapes. Some of these sites is abandoned. The choice of classes was caused by the low diversity of crops in the study areas. Most of the cropland is either abandoned or planted with soybean. Even the share of oats, a strategically important green manure for soybean cultivation, is small. Other crops (buckwheat, timothy grass) that grown in the Khabarovsk District are practically not sown in the studied areas of the Amur region. These three classes were chosen on the basis of known ground reference data in both areas.

Round 2
Reviewer 1 Report
The authors did not make enough modifications with respect to our comments. Most of the time, the authors responded to our comments but they did not try to improve their paper. This is not acceptable and they should make changes on their paper, taking our comments into account.
Like they have responded to “What was the package used for raster data handling in Python?”. But that is absent in the manuscript.
I could not find where they have mentioned the limitations of the present study.
Please properly mark the changes made in the manuscript.
Reviewer 2 Report
The author don't respond point by point to the suggested changes.